# Making Climate Risks Governable in Swedish Municipalities: Crisis Preparedness, Technical Measures, and Public Involvement

**Rolf Lidskog *** and **Linn Rabe**

Environmental Sociology Section, School of Humanities, Education and Social Sciences, Örebro University, SE-70182 Örebro, Sweden; linn.rabe@oru.se
* Correspondence: rolf.lidskog@oru.se

**Abstract:** Creating preparedness for climate change has become an increasingly important task for society. In Sweden, the responsibility for crisis preparedness rests to a large extent on the municipalities. Through an interview study of municipal officials, this paper examines municipalities' crisis preparedness for climate change and the role they assign to citizens. The theoretical approach is that of risk governance, which adopts an inclusive approach to risk management, and that of risk sociology, which states that how a problem is defined determines how it should be handled and by whom. The empirical results show that the municipal officials mainly discuss technically defined risks, such as certain kinds of climate-related extreme events, the handling of which does not require any substantial involvement of citizens. Citizens' responsibility is only to be individually prepared, and thereby they do not require municipal resources to protect their own properties in the case of an extreme event. The municipalities, however, feel that their citizens have not developed this individual preparedness and therefore they try to better inform them. This analysis finds five different views of citizens, all with their own problems, and to which the municipalities respond with different communicative measures. By way of conclusion, three crucial aspects are raised regarding the task of making societies better prepared for climate change.

**Keywords:** climate change; cloudburst; crisis preparedness; extreme weathers; heatwaves; public involvement; risk communication; risk governance; Sweden

## 1. Introduction

Creating preparedness for climate change has become an increasingly important task for society to handle. As stated in the recently launched IPCC report on impacts, adaptation and vulnerability, global climate change has caused widespread adverse impacts and related losses and damage to nature and people [1]. Extreme weather is increasing in frequency. Heat waves, intense rainfall, drought, and tropical cyclones are more common. Cities are hotspots for these impacts, partly because more than half of the world population lives in cities, partly because many cities, not least their infrastructure, including energy and transport systems, are more vulnerable. Cities are being increasingly adversely affected by extreme weather such as heatwaves, storms, drought, and flooding—but also from slow and creeping changes, such as sea level rise.

In the European Union climate adaptation has developed to become a prioritized area on parity with climate mitigation. The European Commission adopted in February 2021 a new strategy on adaptation to climate change [2]. The new strategy sets out how the European Union can adapt to the unavoidable impacts of climate change and become climate resilient by 2050. This means that the EU should not only be climate neutral by 2050, but also resilient to the impacts of climate change. The new European Climate law, which entered into force in July 2021, commits the EU and its member states to continuously making progress in promoting adaptability, strengthening resilience, and reducing vulnerability

to climate change [3]. Thus, all member states must develop policies and measures for climate adaptation. Most recently, the IPCC [4] found that for Europe, there are several adaptation options and barriers. Among the options are behavioral change combined with urban planning and building interventions (for heat), as well as land use changes and reserving space for water and ecosystem-based adaptation (for flooding). Among the barriers are a lack of political leadership, a lack of urgency, resource limitations, and a lack of engagement from the private sector and citizens. The report finds that adaptive planning is still limited across Europe.

Sweden, an EU member state widely considered a forerunner in environmental policy [5,6], intends to become the world's first fossil-free welfare state. However, it is not only for climate mitigation that it has high ambitions, but also for climate adaptation. In 2018, the government adopted a climate adaptation strategy, the objective of which is to "develop a long-term sustainable and robust society that actively meets climate change by reducing vulnerabilities and seizing opportunities" [7]. The responsibility for safety and crisis preparedness rests to a large extent on the municipalities.

The distinct local character of climate adaptation makes it fundamentally different from the scale of required mitigation efforts. It is always particular places that are affected by storms, heatwaves, cyclones, and flooding. What is affected is also to a large extent determined by local planning and the capacity of local governments to prevent and handle disasters.

In Sweden, the municipality is central in the work of adapting society to climate change and creating preparedness for climate-related crises [8]. Preparedness in this context means to be in a state of readiness, to have resources available and to be organized to use them [9]. The municipality is responsible for physical planning, water and sewerage, care, and rescue services, which are activities that are affected by a changing climate. It is responsible within its borders for the work to prevent and reduce the consequences of natural disasters, where physical planning plays a major role [10] (p. 97). According to national legislation, municipalities must assess the risk of damage to the built environment caused by climate change, for example, damage caused by flooding and erosion. Furthermore, the Swedish civil contingencies model encourages the municipalities to strengthen citizens, private actors, and civil society in their preparedness [11]. The municipalities are thus central to the work of both preventing and reducing climate-related damage, as well as in activating the public regarding crisis preparedness and supporting them in taking their own responsibility for it. Even so, few empirical studies have explored the interactions between citizens and municipalities in Swedish climate risk management and adaptation [12].

At the same time, the municipality is a complex organization that is responsible for many areas, and which often runs within strict economic limits [13]. This means that it must handle many issues, among which climate adaptation is only one. Often, municipalities must balance and navigate between competing objectives, and where climate-related risks may not be prioritized in favor of other, more manifest, issues. This may explain a finding in a recent evaluation of municipalities' climate adaptation work whereby the majority of municipalities had identified the need for various climate adaptation measures within their municipalities, but that this had not been followed by sufficient implementation [14].

This complex situation forms the background for this paper. It focuses on how municipal officials assigned to handle risk and crisis preparedness understand and prepare for climate-related risks, including how they understand their citizens' perception of and role in climate risk preparedness. (We use here a broad understanding of citizens, which is partly interchangeable with residents. Citizen concerns the formal relation to the nation-state, including rights and duties. Resident concerns where a person lives. A municipality can be host for many people that are not residents—such as commuting employees, temporary visitors, and tourists—and not citizens—such as people younger than 18 years, people with foreign citizenship, and undocumented migrants). It is of great interest to investigate a country with high national ambitions regarding climate adaptation, where municipalities have a large responsibility for handling this area, and where national authorities support

this work while also emphasizing citizens' responsibility for their own preparedness [15]. This study is explorative, aiming to investigate the character of a problem rather than draw substantial conclusions. The aim is to explore how municipal officials define, describe, and conceptualize the issue of local climate preparedness and the role of citizens in this work.

An interview study of municipal officials is conducted. The guiding research questions are: Which social definitions do the municipal officials develop around the issue of local climate risk preparedness? How do they understand the citizens and their role in developing local climate preparedness?

This paper is structured in five sections, this introduction included. The second section develops the theoretical approach—that of risk governance and risk sociology. The third presents the study design—case selection, materials, and methods. The fourth section presents the results, how municipal officials understand and prepare for climate-related risk, including how they perceive the citizens' perception of and role in climate risk preparedness. The fifth and concluding section puts these results in a wider context and discusses their implications for long-term preparedness.

## 2. Theoretical Framework: Risk Governance

In modern society, risk has permeated almost all aspects of social life [16,17]. There are now few, if any, social sectors where thinking in terms of risk is absent, and most organizations, from the local to the global level, have to consider and handle risks. Society seems to have no option but to organize itself in terms of risk, which implies that regulation and policymaking can to a greater extent be understood as risk management [18–20].

The risk panorama has also changed and broadened. Whereas previously the focus was mainly on certain products (e.g., harmful substances and technical artifacts) and systems, a growing number of risks concern how actors act upon what they see as risks [21]. Public agencies, private companies, and interest organizations now include broader questions of both how organizations and people act upon perceived risk and how they evaluate the credibility and legitimacy of the organizations that handle these risks [22]. For example, if there is a public perception of a coming crisis, this may lead to a negative spiral of increasing hoarding and bunkering of certain kind of goods. Additionally, if people mistrust the capacity of a public agency to prevent and handle risks and crises, they may act in a way that counteracts the work of that agency. The work of the public agency can also be indirectly counteracted by people who raise far-reaching claims that the agency cannot meet, and the agency thereby needs to allocate resources to preventing or handling public outrage. Thus, public perceptions can be a source of risk in the sense that these perceptions can pose a threat to the legitimacy and stability of existing ways of managing risk [21,23]. A public agency, as well as other regulatory organizations, therefore needs to consider what the public thinks and feels about a risk, irrespective of whether it shares these understandings. A large body of research literature explores the factors that affect the public's perception of risk, as well as the factors affecting their understanding of the authorities handling these risks [24]; however, less scholarly attention has been given to the authorities' perception of citizens and how this may affect their actions [25].

*Risk governance* was developed as an effort to understand and handle the complex situation of risk [26–28]. It refers to the various ways in which diverse actors—public and private organizations, as well as interest organizations and individuals—deal with risks surrounded by uncertainty, complexity, and/or ambiguity [29]. It includes all parts of risk regulation—assessment, management, and communication—but these are not conceived as discrete activities that follow on from each other, but rather as parts of a dynamic process where different actors interact and negotiate to define and regulate risks [30]. It thereby also goes beyond traditional ways of regulating risk by stressing that there are many legitimate ways to define and evaluate risks, and therefore stakeholders and the public have something to contribute. Furthermore, the traditional top-down way of managing

risk is increasingly seen as obsolete, with governance being seen as a more feasible and efficient way to handle complex, ambiguous, and contested risks.

However, risk governance is not a manual for how to regulate complex risks, but an approach highlighting what should be considered when regulating uncertain, complex, and maybe also contested risks [29]. Regulation should be more inclusive, in terms of actors (including the public, but also other stakeholders and NGOs) and forms of knowledge (incorporating broader views of risks than are provided by technical risk analysis). It should also be transparent and communicative, allowing organizations and citizens (irrespective of their possible involvement in the regulatory process) to be able to evaluate a regulation or a regulation's purpose. Thereby, it is hoped that the regulation will be more robust, that is, legitimized and trusted by the public, appropriate to the targeted risk, and effective in implementation.

Risk governance is widely held by researchers and many organizations as representing a way forward. However, there are also critical voices. Even regulatory bodies, which on the surface may have embraced such an inclusive approach to risk regulation, may not in practice have changed their ways of governing risk [31]. An inclusive approach can be costly and time-consuming, it may not lead to shared agreements, but rather increase divergence in views about what needs to be done. Furthermore, inclusion may also have an implicit strategic function, spreading accountability if an adopted regulation is later shown to be inadequate [32]. Furthermore, it can also provide regulators with better opportunities to influence public opinion, serving as a form of steering activity [33–35]. All governing activities include work to shape, negotiate, and spread the meaning of a particular issue or situation, including what social definition of reality counts, what knowledge is authoritative, which experts are trustworthy, what intervention is legitimate, which rules should be followed, and what temporal-spatial frame is appropriate [20].

This criticism echoes the perspective provided by risk sociology, which stresses that risk is always situated in a particular social context and necessarily connected to actors' activities and framings [17,36]. A *technical risk framework* understands risk as the product of the probability and consequences of an adverse event, which are possible to measure through scientific instruments, and predictive models can be elaborated to take action in order to minimize risks—either to reduce their likelihood or mitigate their consequences [27] (pp. 12–19). Central to this understanding is that, whereas professional experts' measure of risk is objective and correct, public perception of risk is subjective and incorrect. In contrast to this perspective, *risk sociology* stresses that there are many ways to frame, define and evaluate risks [37]. Thus, risks are here seen as social phenomena, where institutions, organizations and people use their beliefs, values, and knowledge to single out what is risky and what is safe, thereby making it possible to navigate in the world [38]. The focus is here on how risks are made knowable and governable.

If, for example, an actor adopts a technical risk framework, it means that a phenomenon is singled out as a risk, and an institutional machinery of risk assessment and risk management is activated to define and handle this risk. Other understandings of risk may flourish, socially amplified by, for example, rumors and the media [39]. That there are other understandings of risk does not imply that the original definition of the risk needs to be revised, but only that these understandings must be considered as a political reality—they may hinder proposed measures or result in loss of legitimacy if implemented. If, on the other hand, a more open and constructive view of risk is applied, it would mean that all understandings of risks are important per se, they all have something substantial to say about a risk and how it should be governed.

Important to note is that risk sociology does not imply a far-reaching relativism [37]. It only states that, contrary to a technical risk framework, risks are never independent of their context and how they are framed. To regulate a risk is not only about setting limits and developing measures, but also about framing the risk as such, which also includes deciding which actors are of relevance in the handling of this risk (defining their roles, mandates, and responsibilities). Thus, a risk governance approach, fertilized by risk sociology, means

that it is of great interest to investigate how an abstract and multifarious risk such as climate change is handled in practice.

## 3. Research Design, Method and Material

The Swedish municipalities that have come furthest in their climate adaption work have often been those affected by extreme weather events and are often larger in size [10] (p. 22). Studies have also shown that national risk communication prioritizes the urban population [11], probably because cities are more complex, and their citizens are more dependent on different technical infrastructures. At the same time, the research literature exploring citizens' preparedness often does so in small-scale or rural settings, with an assumption that these populations are easier to activate in a disaster [24], and urban areas are less explored. In this study, two municipalities were selected: one metropolitan area (Stockholm city, the biggest city in Sweden, 1,000,000 citizens) and one large city area (Örebro municipality, the sixth-biggest city in Sweden, 160,000 citizens). Besides being urban areas, another selection criterion was that they had experienced nationally acclaimed crises in recent times. The two municipalities were not chosen for comparative reasons but rather in order to gain a broader empirical material on how municipal officials understand climate-related risks. Therefore, no detailed information is given about characteristics of the municipalities.

The design of this study is that of an exploratory case study [40–42]. By selecting a country with far-reaching ambition for climate adaptation and preparedness, and where the municipalities are seen as crucial in this work, information can be gained about a task that is still in its infancy in many other countries. This does not mean that these results can easily be transferred to other contexts [43], but nevertheless, important insights can be gained for other municipalities and states working with climate preparedness.

This study focuses on professional expertise—employees with the competence, experience, and mandate to conduct or suggest relevant interventions for an organization [44–46]. Professional expertise is needed not only to address a problem but also to define it [47]. As most environmental problems are complex and ambiguous issues, they need to be demarcated to render them governable. All problem solving is based on a particular level of understanding that makes it possible to imagine a problem, analyze it, and propose a relevant solution [48,49]. By studying the expertise of professionals working with climate preparedness, knowledge is gained on municipal work with climate risks.

Interviewees were identified on the basis of their responsibility for and active part in municipal work regarding climate-related preparedness and adaptation. The study searched for a broad representation of interviewees, thereby gaining a general view of how climate risks are defined within the municipal administrative organization. This focus implied that municipal communicators were not included in the study (which focuses on the work aiming towards climate preparedness, and not how it is communicated to target groups). However, two communications officials were recommended by other interviewees due to their important role in the overall organization of risk management. Altogether, twelve interviews were conducted with fifteen municipal officials (one of the interviews was conducted in a group of two and one in a group of three, because the interviewees suggested they would complement one another) responsible for climate-related preparedness in their municipalities. Three interviews were conducted with officials at central management (strategy for security and communication), three with environmental management, five with district management/task units (technical support, welfare, care and transportation) and one with municipal companies (water and sewer). All had a specific responsibility linked to preparedness, risk prevention and/or climate change adaptation, but about half of them worked on these issues for 10% or less of their working time. Of the interviewees, nine had long experience working with the municipality (at least six years, often more than ten years), while four were new at their current position. Of the fifteen respondents, twelve were women and three were men.

The interviews were semi-structured. An interview guide was used to introduce themes that allowed the interviewers to ask follow-up questions and expand on topics that emerged during the interview [50]. This method was selected for its proven suitability for unfolding argumentation and gaining an inside perspective on interviewees' understanding and reasoning.

The interview guide (enclosed as Supplementary Material) centered around themes of risk perception and assessment, the role of citizens, the division of responsibilities, risk and vulnerability assessment, and effect/outcome. The interviews focused on climate change, but to illustrate and develop their reasoning, interviewees often used other experiences, for example the ongoing COVID-19 pandemic. The interview study was conducted between September and October 2021. Because of the ongoing pandemic at this time, all interviews were carried out digitally (through Microsoft Teams). The interviews were recorded and transcribed verbatim. A contextualized thematic analysis was conducted [51] using Nvivo software to analyze the qualitative data. In the analysis, special attention was paid to thematizing everything that was said about risk perception, risk management, and the perceptions of citizens. The analysis of the material was carried out hermeneutically, whereby the content of a particular code was constantly related to the entire content of the interview.

## 4. Results: Transforming Abstract Climate Risks to Governable Entities

In the following, we present the results from our analysis of how municipal officials understand and prepare for climate-related risks. This is structured in four parts; first, we describe how they perceive climate-related risks, followed by how they perceive the citizens' preparedness for this kind of risk. Thereafter, we investigate the level of trust municipal officials put in risk information as a tool to increase their citizens' climate preparedness. The analysis reveals that the municipal officials have a differentiated view of citizens. In the last part, we construct a table that summarizes these views.

### 4.1. Perception of Risk

Climate-related risk is a rather vague and abstract notion, and it must be made relevant in local and context-specific terms to support crisis preparedness in a municipality. Therefore, it is of great interest to investigate what meaning local officials attach to climate-related risks, what kind of preparation they find relevant and how they interpret other understandings of climate risk.

A representative quote from central administration explains climate-related risks such as this:

> "Generally, we do risk assessments based on probability and consequences of an event. But it doesn't matter what caused the event. It may be an antagonist attack or a cloudburst. It's still a process or activity that needs to function regardless of external disturbances such as heat and water. Climate change will affect the frequency, scale, and magnitude of disturbances. The cloudbursts are heavier and hit us more often. But the risk itself is the same" (*Security strategist*, *central administration*, *Stockholm City*.)

This quote illustrates the "all-hazard approach" that gives the basic logic for Swedish civil contingency [11,52]. When exemplifying what climate change-related risks their municipality is facing, the interviewees' responses focused on weather events that could be forecasted, especially rain fall and heat waves. Climate change was generally argued to intensify existing risks, but not thought to create new complexity or add ambiguous and contested risks. The intensification of weather was rather perceived as measurable and relatively predictable. Some interviewees even argued that the likelihood of climate change-related risks of extreme weather sets it apart from other threats faced by Swedish municipalities. Whereas other identified risks (such as cyberattacks, deliberate acts of senseless violence, large transport accidents, or industrial fires) indeed would affect the activities of the municipality and, if reoccurring, even threaten the stability of the local

governance system, those risks are not perceived as imminent. Furthermore, their experience of such events was limited, and to some level the risks were perceived as unknown. In contrast, all interviewees had experience of extreme weather, particularly cloudbursts and heat waves, and were convinced that their municipality would experience more such climate-related events in the future. The interviewees seemed to set climate change apart as a risk, since it is neither random nor are its consequences completely uncertain. Hence, these consequences of climate change can, and should, be planned for, as illustrated by this quote from a traffic planner:

> "We gonna see these heavy rains more often. The kind of rains we previously saw every hundred year or so will be frequent and we have to adapt to it. [ . . . ] It's a bit slow with resources, the politicians struggle to prioritize risks before already demonstrated issues, but we are talking about it and we have a new position—a cloudburst strategist. [ . . . ] We do what we can, for example we have projects to make extensive water run into lakes and parks [instead of undersized draining systems and buildings]. [ . . . ] I understand that we need more housing, but when they build in areas with high risks of flooding . . . In thirty years they will realize what a mistake that was." (*Planner*, *Traffic office*, *Stockholm City*.)

This quote is representative of the way in that the municipal officials seemed to define the climate risks they must handle as those related to extreme weather events. These risks are mainly defined within a technical risk framework, and technical measures (such as increased curbs and changes in land sealed) are seen as central measures for preparing to avoid or minimize the consequences of these risks. The technical risk definition is a common, and often very effective, way of dealing with many risks where an institutional machinery of risk assessment and risk management has been activated to define and handle the risk, which then should be disseminated in society. In the planning and management of technical risks, the municipalities can use their expertise, while the involvement of layman opinions seems unnecessary and could even possibly be counterproductive, as illustrated by this quote from a security strategist:

> "You see, [developing risk assessments] is a delicate craft. [ . . . ] To involve citizens in our assessments of vulnerability is very hard to practically manage, and its information we don't want to make publicly known. We rather keep this assignment as experts and don't involve citizens in it." (*Security strategist*, *Central administration*, *Stockholm City*.)

This is complemented by a quote from a security strategist from another municipality:

> "We are not involving citizens in the risk assessment. We could probably be better at involving them, but . . . there are challenges to get good representation in public participation. Some groups are more eager to get involved and it can create bias towards their needs and wants whereas other groups tend to be absent, and their perspective missed out." (*Head of security*, *Central administration*, *Örebro Municipality*.)

Hence, in the municipal process of preparing for climate risks, the citizens have no obvious, active role. Instead, the citizens are encouraged to increase their own individual preparedness.

### 4.2. Perception of the Citizens

There was a common understanding among the interviewees that the municipalities had a responsibility to engage their citizens in crisis preparedness and support them in taking responsibility for it, as shown by this quote:

> "The goal [with the municipalities risk communication] is to make the citizens aware of the risks and dangers that may affect their every-day-life. Make them conscious about the things they must make risk assessments of and have preparedness for. Make them aware of their responsibilities as citizens and responsibilities

of the municipalities. [ . . . ]. In this way a resilience is created in the population, so that actors involved in crisis management can prioritize the most urgent and not put resources on those who should be able to manage on their own." (*Security coordinator, technical support unit, Örebro Municipality*.)

Several interviewees argued that prepared citizens are an aspect of the Swedish civil contingencies model that state agencies cannot otherwise control, as illustrated by this quote:

"It's positive when citizens activate themselves for preparedness. The more people prepare, the less people get affected by a crisis. And so, less people need to turn to the municipality for support." (*Urban planner, municipal district administration, Stockholm City*.)

The ideal citizens seem to be those who are individually prepared to sustain themselves on their own without the support of public services for a certain amount of time, while the authorities focus their resources on resolving the cause of the situation. This means that citizens should be generally prepared for different kinds of threats by storing food, water, and medicine, securing heat/cooling, and make sure to be able to receive crisis information from the municipality and public agencies. Hence, the focus is on the citizens' individual and material aspects of preparedness. However, the general opinion seems to be that the level of preparedness is currently too low, as surmised by this quote:

"No! I must be frank—the citizens do not have a preparedness" (*Security strategist, Central administration, Stockholm City*.)

Numerous unprepared residents claiming public resources for individual protection would make it harder for the municipality to carry out suitable risk management. If the public is not cognitively and practically prepared for a crisis, the municipality will have to allocate resources to manage their concerns and claims, regardless of whether they are reasonable or not. This quote was given to illustrate the problem of the unreasonable use of public resources:

"We clearly see that if you are a native, affluent Swede, you call [the environment and health protection] and whine as soon as something is unsettling. And we put hours and hours into investigating bouncing sounds from the racquetball court. But we rarely hear from socio-economically disadvantaged areas. Why is it so? Because their standard is so high, they don't need the municipality? No, obviously not. It's because they don't know they have rights, and that we can support them [to keep responsible actors accountable]. It's the same thing with elderly at retirement homes. [ . . . ] They don't know they can turn to us." (*Head of unit, environment and health protection, Örebro municipality*.)

However, it also illustrates the attitudes of different groups of citizens and their different relationships with the municipality, as well as gives clues as to why citizens may be unprepared.

Five different reasons why citizens are unprepared emerged from the material: lack of interest, lack of knowledge, lack of agency, mistrust in public institutions, and/or misdirected responsibility.

The five reasons may co-exist with and reinforce each other, but they also illustrate the diverse and heterogenous character of the public. Groups of citizens tend to lean more towards one or the other of these reasons, depending on their position. In relation to crisis preparedness, this seems to differ in three fundamental ways: (i) the citizens have different capabilities to understand municipal information about the need for crisis preparedness; (ii) they have different resources and capabilities to develop crisis preparedness; (iii) they have incorrect beliefs regarding the capacity and responsibilities of the authority in matters of crisis. For example, they may assume that the state and municipality will prepare for and protect citizens from all kinds of consequences in a crisis, and therefore the citizens do not need to develop crisis preparedness on their own.

*4.3. Risk Information: Dealing with Socially Amplified Risk and Troublesome Citizens*

Risk information is presented by the interviewees as the most prominent tool (possibly even the only one available) for the municipality to convey the importance of preparedness to citizens. The most prominent example regarding public information is the yearly MSB-led public awareness campaigns. These campaigns intend to activate citizens' urge to strengthen their households' preparedness in an "all-hazard approach" [52]—that is, to be generally prepared for different kinds of threats by storing food, water, and medicine; securing heat/cooling facilities; and having a way to receive crisis information from the municipality and public agencies.

How the municipal officials see risk communication in relation to the complexity of preparedness for climate risks, with the challenge of making them governable for the municipality and relevant for the citizen, is illustrated by this security team member:

> "The greatest challenge is to make the super complex issue [of climate change] relevant for people. To make them susceptible . . . What we suggest they do in terms of preparation is so basic: "keep cans of food at home". It's a great challenge to frame this kind of abstract risks and link them to actions for risk limitations. [Internally] we talk a lot about new types of crises, creeping crises, as the climate change is. It's not concrete for most people. You cannot grasp the causation. [ . . . ] Suddenly you stand in your basement with water to your knees and you don't know how it happened. It seems so many steps away [from the cause of climate change]. That's one of the great challenges, to make it relevant for people. Such a challenge, I don't envy the communication officials who must deal with this." (*Security strategist*, *central administration*, *Stockholm City*.)

In this aspect, the municipal officials reflected on how different social processes may substantially influence the municipal work for climate preparedness if the citizens remain unprepared.

With the technical definition of risk as a backdrop for the discussion, there was often a social "add-on" presented by the interviewees. The significant factor determining the magnitude of an event was argued by many to be how the public reacts on the situation. A significant component of the municipalities' crisis management strategy is to meet the extensive demand for information from the public and the media. In the case of an extreme weather event, the municipalities can expect a steep increase in phone calls, e-mails, and social media activity during and after. Regular, well-functioning citizens will contact the municipality to get information, claim resources, and demand accountability.

A smaller number of the interviewees also addressed the challenges of the extensive offerings of voluntary resources during a crisis. These officials argued that the municipalities generally do not have the capacity to utilize the resources offered to them by volunteers (often private individuals), neither do they have the right contacts to collaborate with civil society for climate change adaptation and preparedness. Hence, community building for a robust society is not a conscious strategy, according to the interviews. Rather it is argued that the best way for the municipalities to support other actors in their preparedness is to offer correct and balanced information about risks and the ways in which citizens can individually prepare to sustain themselves in the face of a crisis. The ideal outcome seems to be calm citizens who individually manage their situation so that the municipality can do its job without unnecessary interference. A worst-case scenario would be if the citizens panicked and acted in a suboptimal way:

> Interviewee 1: "If the citizens have a low level of preparedness, [the municipality] must put resources into making sure that not every grocery store is looted, and people start to fight each other for toilet paper. Such things would steal attention from what we really must do. Like making sure that the elderly in our care at retirement homes still get what they need."

> Interviewee 2: "Indeed! Also, people who are unprepared are erratic. They may escalate the situation and cause new types of problems, like if people hit the

streets to protest and it gets messy, and the police need to maintain order, and a lot of things can go wrong … It means tons and tons of public resources on the wrong thing." (*Planner and head of department, environment, and health protection, Örebro municipality.*)

Even if only the two group interviews pushed the scenario as far as public panic and riots, several interviewees were concerned about the public's level of cognitive preparedness and what social masses caught off guard might do in the face of danger. Some municipal officials reflected on "the right amount" of information, as "too much" transparency may create unnecessary ambiguities, causing worries and fueling unwanted behavior among citizens. They were concerned that some actors could aggravate the situation by using a narrative that creates uncertainty and worry. This could be the deliberate spreading of disinformation or questioning the credibility of public institutions and expertise-based sources. Apart from antagonists, some interviewees mentioned "preppers" in this respect, as they were perceived to be motivated in their preparations by a perception of exaggerated risks of major cataclysm and distrust in public systems' ability to handle these risks.

However, in general, the interviewees wanted to downplay citizens' worries and emotions as an aspect in risk assessments. When asked if citizens' worries affected the work of municipal officials, all interviewees stated that it did not. In addition, if so, they argued that balanced risk information was able to deal with this kind of problem. Even so, the municipalities want to make their citizens feel safe, so officials try to provide objective solutions on citizens' subjective problems. For example, if citizens say they feel unsafe in an area with a lot of graffiti, the municipality does a clean-up.

The interviewees did not feel that officials should deal with value discussions and make priorities among conflicting goals or interest groups. This was rather thought to be up to the political leadership of the local government. Some of the interviewees indicated that politicians were sensitive to worries expressed by vocal groups of citizens and made actively responding to these concerns a higher priority than the professional judgement of officials and the original plan. Furthermore, politicians are impressionable if the media or political opposition get involved. Several interviewees argue that these kinds of demands from certain, powerful groups of citizens are based on a misperception on public services. Even if the municipality officials may disagree with a subjective or emotional change in priorities on the back of politicians' responses to demanding citizens, they support the political leadership of the municipality as an institutional part of Swedish democracy.

*4.4. A Differentiated View on Citizens*

To secure the national strategy of civil contingencies in relation to climate-related risks, the municipal officials argued that the citizens need to take an active part in their individual preparedness. The municipal officials framed the problem as being mainly one of communication: to get citizens informed about their responsibility to prepare for climate risk. The material shows that the municipality faces different challenges regarding risk information depending on the agency, interest, and rationality of different groups.

The material in this study shows a differentiated view on citizens. This differentiated view is mainly based on how citizens relate to and understand (ignore or amplify) climate risks. Table 1 summarizes the views of citizens that our empirical analysis found. It refines our findings and has an ideal-typical character—that is, it is derived from our empirical material, but we have selected and accentuated certain elements [53,54]. It is thereby not an average list of common features of real-world phenomena, but rather a constructed purification. The table consists of five different categories which constitute different challenges for the municipality. The categories are not even in size, the average citizens constitute most of the population, and thereafter, the categories are presented in descending order depending on scope.

**Table 1.** Municipality officials' different views on citizens.

|  | Average Citizen | Vulnerable Citizen | Disadvantaged Citizen | Demanding Citizen | Prepper Citizen |
|---|---|---|---|---|---|
| **Problem** | Lack of interest | Lack of agency | Lack of kn006Fwledge | Lack of responsibility | Lack of trust |
| **Cause** | Public ignorance | Vulnerable position | Socioeconomic conditions, language deficits | Misperception of public services | Misperception of risk |
| **Agency** | Do not use it | Lacks it | Cannot use it | Uses it wrongly | Uses it wrongly |
| **Solution** | Risk information | Municipal support | Target group information | (Balanced) information | (Balanced) information |

**The average citizen** has the capacity to be informed about their responsibility for preparedness and the capability to take it. Most members of this group, according to the interviewees, are unaware or even ignorant. They have far-reaching expectations that society will assist them in case of need. If this group took responsibility, the municipality could more easily allocate resources to where they are most needed. However, in the case of a crisis, this category is also a resource by virtue of supporting and helping other people.

**The most vulnerable citizen** consists of citizens that are already in the municipality's care, and for them, the municipality oversees all preparedness. These citizens are not at all expected to be informed or involved in national strategies for civil contingencies. Some of the officials dealing with this group almost find questions about their preparedness provocative or ridiculous, as they are so heavily dependent on the municipality for their day-to-day care.

**The disadvantaged citizen** consists of people that are perceived to lack resources and know-how about the Swedish system. They are often clustered in "socially disadvantaged areas" on the urban periphery. The officials believed that this category had less trust in authorities. This group is more exposed to danger, and due to socioeconomic conditions is more vulnerable than the population in general. Adaptation of message (e.g., in easily read Swedish or translated to target group's mother tongue) or direct oral information from a trusted source (e.g., local ambassadors), is used to inform this group.

**The demanding citizen** consists of citizens who claim that the municipality (and the state) should allocate resources to prevent them, and their properties being affected. They express those expectations loudly, often in media, and have access to influential networks. Sometimes they manage to influence the political will to adjust according to their preferences.

**The prepper citizen** consists of citizens who take their individual preparedness very seriously. Even if they could potentially sustain themselves independently without public support for a long time, they are perceived as a problem, since they exaggerate the risks and mistrust the authorities' capacity to handle crises. They are unwanted, as they are perceived to cause anxiety and even panic among other citizens.

Obviously, this is a simplification, but even so, some concluding comments can be made: not all citizens are perceived to have agency in risk preparedness. As the most vulnerable citizens lack agency, the municipality needs to compensate for this with active support in case of a climate crisis. Other citizens have agency, but either cannot use it (disadvantaged citizens), do not use it (average citizens) or use it in a suboptimal way (demanding citizens, prepper citizens). Depending on the municipal officials' views on citizens, they proposed different measures on how to handle this situation. Other than those without agency (vulnerable citizens), the solution is to get the citizens better informed. Most of the interviewees considered the citizens to have equal access to information and almost equal access to preventive measures, making ignorance or indifference the major obstacle in citizens taking individual responsibility for preparedness. As one official put it:

No matter if you are in high school or 90 years old, you can do something to increase your preparedness. [ . . . ] At least you can follow the weather forecast. (*Communication strategist*, *central administration*, *Stockholm City*.)

Irrespective of who you are, almost all citizens have the capability for some basic preparedness that would improve their chances of sustaining themselves without the need/urge to call on public resources due to an extreme event.

## 5. Discussion

This paper focuses on how the municipal officials assigned to handle risk and crisis preparedness understand and prepare for these risks, including how they understand citizens' perception of and role in climate risk preparedness. As shown above, a technical risk framework dominates the understanding of climate change and the municipalities' efforts to strengthen preparedness for climate-change-related risks. As the municipality officials are used to calculating risks on the basis of probability and impact, certain types of climate risks emerged as being more prevalent. Most interviewees identified floods through cloudburst and heat waves as the prioritized risks. These extreme weather events are neither perceived as being random, nor are their consequences completely uncertain, in theory giving the municipality a good chance to plan for them. However, in this planning, the citizens are not perceived as being a counterpart, but mainly as a possible complication that the municipality, through different communicative measures, must handle.

Based on these findings, we would like to continue this exploration by drawing attention to three more general and crucial aspects related to climate change preparedness, and finally some general considerations regarding the wider task of making society better prepared for climate change.

First, this study shows a municipal focus on acute and "traditional" risks when it comes to preparedness for climate change. Climate change is not presented as a complex and integrated creeping crisis, evolving over space and time with unpredictable manifestations, as often described by researchers [55]. The technical definition of risk is understandable in terms of making risk governable, as it implies a need for the reduction in an issue's complexity [49,56]. At the same time, to reduce the complexity and multifariousness of an issue is a risky strategy: it may lead to some important and severe risks not being prepared for, and if they occur, the municipality will have developed no capacity to cope with it. It may also cause the municipality to overlook possible overlapping risks, which can lead to a so-called "perfect storm" [57]. This prompts the question as to whether contemporary risk assessment tools are a proper fit for the creeping crisis of climate change. Thus, the trade-off between well-defined risks and vaguely defined and complex risks should always be kept alive.

Secondly, the municipality has a somewhat contradictory understanding of the citizens and their role in preparedness for climate change. The officials seem largely to perceive the citizens as being passive and without cognitive or practical preparedness for extreme events, yet they have high expectations of them to act responsibly and on their own in the face of a crisis. This perception among officials may have consequences for the development of relationships with and involvement of the citizens. One such consequence may be a blind spot towards social mobilization among citizens: there seems to be an individualistic understanding of the citizen in terms of stressing the individual citizen's responsibility without paying attention to the role of social mobilization and the engagement that civil society has been shown to have during crises [58]. The individualization of citizen preparedness in the face of an abstract and complex risk like climate change may hinder norms and support systems that could otherwise encourage preparedness. Thus, it is crucial not to stress individual responsibility at the expense of the role of social mobilization among citizens.

Thirdly, a warning should be made not to overstate the capacity of risk communication to create a change in behavior. To tackle the problems of low preparedness among citizens, simply providing more information is not the solution [59]. Instead, there is a need to understand citizens' motivations, beliefs, and attitudes in order to build preparedness on the basis of existing cultural values and to design activities as part of every-day practices to make citizens an active (and self-efficient) part in climate change preparedness [60,61]. The lack of citizen involvement in defining and planning for climate change may have an



effect on the legitimacy of crisis management and authority in cases when creeping crises transform into concrete and manifest threats. Thus, to develop local crisis preparedness, there is a challenge to not only develop initiatives and measures for strengthening it, but also to more deeply reflect on how this work frames the issues to be handled and the role of citizens in this work.

Last but not least, what further broad implications for the work on local climate preparedness can be drawn from this study? Obviously, it is hard to draw general conclusions from case studies [40,42]. Much of our findings are context-dependent and are not possible to transfer to other national contexts with their own climate risks, regulatory regimes, and political cultures. At the same time, there is something to learn from other cases. In the latest IPCC report on impacts, adaptation and vulnerability, cities were found to not only be hotspots for impacts and risks, but also for opportunities [1]. Cities are a crucial part of advancing climate resilience development. In this work, the report emphasizes the importance of involving the public in the task of increasing settlements' adaptive capacity. By involving different categories of people in planning, this task can both be more robust and have a greater legitimacy. This means that even if there are technical solutions—technical climate adaptation measures—available for (some) climate risks, municipalities must not ignore the importance of citizens being involved in planning for making society more climate resilient. The challenge is thus not only communicative, but also participatory in that it aims to invite meaningful public participation.

**Supplementary Materials:** The following supporting information can be downloaded at: https://www.mdpi.com/article/10.3390/cli10070090/s1.

**Author Contributions:** Conceptualization, R.L. and L.R.; methodology, R.L. and L.R.; interviewing and data analysis L.R.; writing—original draft preparation, R.L. and L.R.; writing—review and editing, R.L. and L.R.; project administration, R.L.; funding acquisition, R.L. All authors have read and agreed to the published version of the manuscript.

**Funding:** This research was funded by Swedish Civil Contingency, grant number 2020-09584.

**Institutional Review Board Statement:** The study was conducted in accordance with the Declaration of Helsinki. Ethical review and approval were not required for the study on human participants in accordance with the national legislation and institutional requirements (according to Swedish legislation, ethical review is only demanded for studies involving personal sensitive information. This interview study concerns professions (municipal officials) and do not cover any personal sensitive information).

**Informed Consent Statement:** All subjects gave their informed consent for inclusion before they participated in the study. The oral consent was recorded.

**Data Availability Statement:** Not applicable.

**Acknowledgments:** We are grateful for the constructive comments from Mats Eriksson and Hogne Sataoen, Media and communication studies at Örebro University, Sweden, and Anna Olofsson, Risk and Crisis Research Centre at Mid Sweden University.

**Conflicts of Interest:** The authors declare no conflict of interest. The funders had no role in the design of the study; in the collection, analysis, or interpretation of data; in the writing of the manuscript; or in the decision to publish the results.

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
