# Peer review of "Making Climate Risks Governable in Swedish Municipalities: Crisis Preparedness, Technical Measures, and Public Involvement"

_climate, doi:10.3390/cli10070090_

Round 1

Reviewer 1 Report

This paper is well structured and written. A couple of comments are provided as follows.

1.       The reference for section 3 is limited. Examples with details for research design, method, and material are needed.

2.       In the abstract, it mentions “The empirical results show that the municipal officials mainly discuss concrete risks, such as climate-related extreme events, and these tend to be technically defined risks, the handling of which does not require any substantial involvement of citizens.” However, in the manuscript, the authors did not provide enough information to introduce how to prepare the concrete risks such as climate-related extreme events. And why do municipal officials focus on concrete risks?

Reviewer 2 Report

This study offers an interesting glimpse on the municipality officials perspective upon the climate-risk governance in two cities (Stockholm and Örebro, Sweden). The language is good and the background upon the topic is sufficient.

On the other hand, I think this study could have been more structured on the methodology as it appears that some of the conclusions and comments made by the authors are biased.

Some questions I have are below:

Line 220: "Additionally, the two municipalities are at the same latitude, which means that they both have a temperate continental climate."

This phrase should be reformulated as latitude is not necessarily the main decisive factor in climate (e.g. London – daily means of 11°C while in St. Anthony – Newfoundland, Canada, the daily mean is almost 0°C - both cities are on 51° N). Also, some general climate information for Stockholm and Örebro (like a table with multiannual precipitation / mean temperature) would be recommended.

Line 247: "An interview guide was used that allowed the interviewers to ask follow-up questions and expand on themes that emerged during the interview".

Did you have a specific list of questions when you interviewed the officials? I feel like the questions asked and the answers received are too vague for the readers. If available, please provide these in the article.

Line 458: "Table 1 summarizes the views on citizens, consisting of five different categories which constitute different challenges for the municipality". I think this classification (of the citizens) might need more explanations; for example, did the respondents identified a problem, cause etc. for each citizen category or was it an overall simplification made by the authors based on the feelings/thoughts the respondents gave when they answered the questions.  

Some general questions:

Did you remark any differences on the opinions/views expressed by the officials based on the city they were from? Like for example, are the officials in Stockholm more worried about some climate disasters than the ones in Örebro?

Reviewer 3 Report

This is a very good paper, and I would like to see this paper get published. Some minor comments on the paper are as follow:

1.     While in the introduction section, the background of this research is well stated, in the section 2, it will be good to have some more literature review, especially on citizen perception, citizen local government interface and local governance role. 

2.     In the methodology part, since the paper focuses more on qualitative interview, it will be good to know a bit more on the details of the questions asked, and how the responses were validated. It will be good to see some reference to the IPCC report, Working group 2, Europe Chapter. 

3.     Discussion part can be strengthen with specific implantable recommendations for the local governments. 
